# Weak Radiofrequency Field Effects on Chemical Parameters That Characterize Oxidative Stress in Human Fibrosarcoma and Fibroblast Cells

**DOI:** 10.3390/biom13071112

**Published:** 2023-07-13

**Authors:** Hakki Gurhan, Marek Bajtoš, Frank Barnes

**Affiliations:** 1Department of Electrical, Computer, and Energy Engineering, University of Colorado Boulder, 1111 Engineering Dr 425 UCB, Boulder, CO 80309, USA; barnes@colorado.edu; 2Department of Electromagnetic and Biomedical Engineering, University of Žilina, Univerzitná 8215/1, 010 26 Žilina, Slovakia; marek.bajtos@feit.uniza.sk

**Keywords:** weak RF fields, oxidative stress, mitochondrial mass, cell growth, cell death, iron sulphur clusters, radical pair mechanism

## Abstract

In the last few decades, evidence has surfaced that weak radiofrequency (RF) fields can influence biological systems. This work aims to improve our understanding of how externally applied weak RF fields alter concentrations of chemical parameters that characterize oxidative stress. We conducted a series of experiments to investigate the effects of applying weak RF magnetic fields within the 3–5 MHz region on mitochondrial respiration in both human fibrosarcoma and fibroblast cells over a period of four days. Our experimental data show that RF fields between 3 and 5 MHz were able to change the modulation of mitochondrial signaling by changing the cell growth, mitochondrial mass, and oxidative stress. Exposure to RF fields at 4.2 MHz significantly increased the mitochondrial mass and oxidative stress in fibrosarcoma cells. There are substantial concerns that extended exposure to weak RF fields can lead to health effects. The ability to control these parameters by external magnetic fields may have important clinical implications.

## 1. Introduction

In everyday environments, the Earth’s magnetic field is approximately 25 to 65 microteslas (µT), which is often considered weak for most practical purposes. Additionally, in residential and public environments, exposure to radiofrequency (RF) magnetic fields from sources such as Wi-Fi routers, cell phones, or wireless communication systems usually range from nanoteslas (nT) to low µT. These values are typically also considered weak. Relatively weak magnetic fields have been shown to modify cancer cell growth rates, the concentrations of reactive oxygen species (ROS), and reactive nitrogen species (RNS) [1,2,3,4].

The objective of our work is to improve our understanding of the mechanisms by which weak electromagnetic fields modify chemical reaction rates, ultimately leading to changes in biological systems and potential health effects. This involves a complex chain of events, starting with the physics of electromagnetic fields modifying the orientation of nuclear or electronic spins. These modifications can then influence chemical reaction rates, which in turn affect the concentrations of important signaling molecules. These molecular changes can further impact the structure of proteins and DNA, ultimately leading to health effects. Biological systems consist of multiple oscillating subsystems operating in parallel, each with its own gain and time constants, and interconnected in a way that electromagnetic field perturbations can have different effects at different times.

The detection and modification of biological systems by weak magnetic fields are now widely accepted, and it is well-established that RF magnetic fields can affect biological systems [5]. Building upon the growing body of research, this paper presents experimental results on the effects of RF fields on both fibrosarcoma and fibroblast cells. Furthermore, it proposes a mechanism through which an RF magnetic field at nT intensity can alter growth rates, oxidative stress, and mitochondrial mass based on the previously discovered phenomenon of the “Radical Pair Mechanism” [6].

ROS are highly chemically reactive molecules containing oxygen, many of which include free radicals. ROS are useful as signaling molecules in mammalians, but they also cause cellular damage if produced in an uncontrolled manner [7]. It is estimated that 1–2% of the cell’s oxygen consumption is converted into ROS [8]. This level of ROS production is essential for the physiological control of a variety of cell functions [9]. Both complexes I and III of the electron transport chain in mitochondria are thought to be the major sites of ROS production, even though other mitochondrial enzymes and other electron complexes can also generate ROS [10]. The mitochondrial network may function as a frequency- and amplitude-modulated signaling system and may be sensitive to physiological variables such as ROS scavenging. This is why the correct balance between ROS and antioxidants should be in place [11].

Superoxide, hydrogen peroxide, hydroxyl radicals and singlet oxygen are examples of ROS [12] that can cause cell and tissue damage and have been implicated in various human diseases, including cancer. During oxidative stress, there is a significant increase in ROS production, which leads to modifications in membrane lipids, proteins, and nucleic acids. Oxidative damage to these biomolecules is associated with aging and a wide range of pathological conditions.

Oxidative stress results from an imbalance in the production of pro-oxidants and antioxidants. Increases in pro-oxidants or decreases in antioxidants can disrupt this balance, giving rise to high levels of ROS. ROS formation is a part of the cascade of events that result from the assembly of a complex electron transport system in the inner membrane of mitochondria, and the mitochondrial electron transport chain is the primary source of ROS.

There are possible mechanisms underlying the effect of RF systems on biological processes and molecules [13,14]. One possible mechanism is the radical pair recombination, which may explain the effects of weak RF fields on HT-1080 fibrosarcoma and dermal fibroblast cells mediated by reactive oxygen species. Radical pairs can form with their spins in either an antiparallel (Singlet) or parallel (Triplet) configuration, and they alternate between singlet and triplet states due to the influence of the hyperfine coupling rate. When the pairs are in the triplet state, they are less likely to recombine and more likely to dissociate into free radicals. Applying RF radiation at the frequency corresponding to the hyperfine splitting can prolong the duration of the pairs in the triplet state, thereby increasing the probability of dissociation and subsequently raising the concentration of free radicals [15]. The resonant effect on radical recombination can be influenced by the background static magnetic field (SMF) through the electron Zeeman interaction [16]. An RF field, in combination with background SMFs, can alter the product yields if it resonates with the energy-level splitting resulting from the hyperfine and Zeeman interactions.

Within mitochondria, the presence of ROS can give rise to radical pairs that participate in redox reactions and potentially contribute to oxidative damage or signaling processes. Complex I, located within the mitochondria, is the largest electron-transfer complex in the respiratory chain and contains a significant number of iron-sulfur clusters. Figure 1 illustrates that various iron-sulfur clusters exhibit hyperfine resonances spanning frequencies between 0.5 MHz and 5.5 MHz [17].

Exposure to externally applied oscillating fields around hyperfine resonances can alter the concentrations of molecules or ions in different energy states. This modulation of signaling molecule concentrations, including ROS and RNS, can occur in various molecular species. Furthermore, changes in hyperfine couplings can affect the strength of hydrogen bonds and, consequently, the redox potential of iron–sulfur clusters. These clusters undergo oxidation reduction reactions, and their oxidized or reduced states exhibit paramagnetic properties. Paramagnetic molecules possess unpaired electrons, resulting in a net magnetic moment that interacts with external magnetic fields. In our study, we investigated RF exposures within the 3–5 MHz range, targeting hyperfine resonances, and assessed the cell growth rates, mitochondrial mass, and oxidative stress as endpoints of interest.

## 2. Materials and Methods

### 2.1. Exposure System

The experiments were carried out in an incubator containing a mu-metal shielding box and a set of Helmholtz coils. A mu-metal shielding box (Figure 2) was placed inside the incubator to reduce the background magnetic fields so that the SMFs and RF fields could be controlled, and reproducible conditions could be obtained. Temperature, humidity, and carbon dioxide variations were also controlled.

Uniform magnetic fields were generated by employing square Helmholtz coils with 30 turns. One set of Helmholtz coils was activated for the exposed cultures; the other served as the control. The control cells in Helmholtz coils were maintained with an SMF of 45 µT, while the treated cells were exposed to a combination of SMF of 45 µT and RF fields at varying frequencies. Both the 45 µT SMF and the RF magnetic fields are oriented vertically to the flasks or well plates where the cells are exposed. Figure 3 provides a visual representation of our experimental setup, including the Helmholtz coils positioned inside the mu-metal cage.

The control and treated cells were separated by a mu-metal shield. The windings of the Helmholtz coils were made of 1.63 mm wire to overcome heating problems and to keep the cells’ temperature close to 37 °C. The dimensions of the coils (side = 15 cm) were chosen taking into consideration the size of the culture flasks and well plates.

In the case of static magnetic field creation, the current flowing through the coil was direct current (DC), and the measured resistance was 0.8 ohms. However, when using the same coil for generating alternating current (AC) fields, the current flowing through the coil was characterized by higher frequencies. At RF frequencies, the skin effect becomes more pronounced, causing the current to concentrate near the surface of the conductor. As a result, the effective resistance of the coil was around 5 ohms, which differs from the measurements made at DC. In the RF experiments, the current injected into the coils had an approximate value of 0.005 amps. In contrast, for the static field experiments, the measured current injected into the coils was 0.11 amps. We have calculated the dissipated power for static field exposure by the Joule effect using the given current of 0.11 amps and resistance of 0.8 ohms. By substituting these values into the formula, we find that the dissipated power is approximately 0.00968 watts (or 9.68 milliwatts). On the other hand, the dissipated power by the Joule effect for the RF experiments would be approximately 0.0125 milliwatts. Additionally, the inductance of the Helmholtz coils was determined to be approximately 15 µH.

The measurements of the SMF were made using the fluxgate magnetometer FGM-4D2N with a resolution of 0.1 µT (Walker Scientific, Worcester, MA, USA). The homogeneity of the exposure fields varied by ±5% across the region where culture flasks and well plates were placed. An electromagnetic field in a range between 3 MHz and 5 MHz was produced by a signal generator (Hewlett Packard, 8656B, 0.1–990 MHz). The RF magnetic field component between 3 MHz and 5 MHz was measured with a passive loop antenna (Beehive Electronics, Model 100C, calibrated sensitivity up to 3 GHz). With the coils unplugged, we measured a DC background field of approximately 0.5 µT inside the mu-metal box. This value represents the residual magnetic field present in the environment. For the AC fields, we measured the noise floor using a highly sensitive receiver. The noise floor readings were in the picotesla (pT) levels, which correspond to the lowest sensitivity of our AC measurement setup. These pT-level measurements indicate the minimum detectable AC field strength in our experimental setup. In our experiments, we measured the induced electric field to be less than 10 Volts/m. This value was obtained by careful measurements and analysis of the electromagnetic field distribution within our experimental setup.

When producing RF magnetic fields using a signal generator and Helmholtz coils, we needed to adjust the amplitude of the voltage to maintain a constant flux density at different frequencies. This is because the relationship between the voltage, frequency, and magnetic field strength is not linear. The magnetic field strength produced by the coils is determined by the current passing through them. In turn, the current is related to the voltage applied to the coils, the coil geometry, and the impedance of the coils. The impedance, in general, varies with frequency. To maintain a constant magnetic field strength (flux density) as we increased the RF frequency, we needed to compensate for changes in impedance. This was achieved by adjusting the amplitude of the voltage applied to the coils. The applied RF magnetic fields during the exposure had an amplitude of 20 nT. At the intensities of 20 nT within the frequency range of 3 MHz to 5 MHz, thermal effects are not expected to be significant. We monitored the temperature changes during the experiments and observed that the maximum temperature increase was less than 0.2 °C. This indicates that the SAR-induced heating was minimal and did not result in significant temperature changes within the cell culture. The presence of a fan inside the incubators and the inclusion of 8 holes, each 26 mm in diameter, in the mu-metal box serve the purpose of circulating air within the incubator. This air circulation helps to maintain a uniform temperature throughout the incubator, ensuring that temperature effects are minimized.

### 2.2. Cell Growth

Experiments were performed with the HT-1080 human fibrosarcoma cell line (ATCC CCL-121) and human primary dermal fibroblast cell line (ATCC PCS-201-012). HT-1080 fibrosarcoma cells were grown on Eagle’s Minimum Essential Medium (EMEM) (ATCC 30-2003) supplemented with 10% fetal bovine serum (FBS) (ATCC 30-2020). On the other hand, fibroblast cells were grown on a fibroblast growth medium (Sigma Aldrich 116-500). The cells were cultured in treated flasks of 75 cm^2^ to expand cell numbers. After reaching a confluence between 70% and 90%, the cells were seeded in other flasks of 25 cm^2^ or 24-well cell culture plates, depending on the type of experiment. We changed the medium for a new fresh medium for both the control and treated cells two days and three days after the start of the experiment to mitigate the impact of the extended culture duration (four days) such as depletion of nutrients and accumulation of metabolic detritus.

### 2.3. Fluorescence Measurements

Fluorescent assay measurements were conducted using an iBidi 24-well plate, with a surface area of 1.54 cm^2^ per well. In each experiment, nine wells were utilized: six wells for cells and three wells for blank subtraction. The middle six wells on two plates, one for the control and one for the treated samples, were seeded at a concentration of 5000 cells/cm^2^ and incubated at 37 °C with 5% CO_2_. A volume of 1 mL of the cell suspension was added to each well to facilitate cell growth and exposure. The selection of six wells located in the middle section of each plate was specifically made to measure fluorescence intensities. This choice was made considering the homogeneity of magnetic fields in this region, ensuring consistent measurements across samples. Within each well, two samples were collected for fluorescence analysis. To ensure the robustness and reliability of the findings, the experiments were independently replicated twice (*N* = 2), resulting in a total of 24 samples. For blank subtraction, three wells were designated and filled with phosphate-buffered saline (PBS) only. These blank wells accounted for background fluorescence and enabled accurate measurements. Fluorescence studies were performed using a multi-detection microplate reader, specifically the Varioscan LUX (Thermo Fisher Scientific, Waltham, MA), after four days of continuous exposure to the SMF and RF fields. Each experimental condition was examined separately, and a total of 24 measurements were taken per parameter. The mean and standard deviation were calculated based on these 24 measurements for each experimental condition.

#### 2.3.1. Oxidative Stress

RF-field-induced oxidative stress was determined by using CellROX Green reagent for oxidative stress detection (Life Technologies, Carlsbad, CA, USA). HT-1080 human fibrosarcoma cells and human fibroblast cells were stained with 5 μM of CellROX Green Reagent by adding a probe to the complete medium and incubating cells at 37 °C for 30 min. The cells were then washed, suspended with PBS and analyzed on a fluorescence plate reader with appropriate wavelength settings (excitation at 507 nm, emission at 525 nm).

#### 2.3.2. Mitochondrial Mass

Mitotracker Green FM probes were used to label mitochondria. This probe passively diffuses across the plasma membrane and accumulates in active mitochondria. A 50 μg Mitotracker product was dissolved in 74 μL DMSO to prepare 1 mM Mitotracker stock solution. Then, a 1 mM Mitotracker stock solution was diluted using 370 mL PBS to achieve a using working concentration of 200 nM. After incubation at 37 °C for 30 min, staining was complete. Finally, the staining solution was replaced with fresh prewarmed PBS and mitochondrial mass was detected using a Varioscan LUX multi-well plate reader (Thermofisher, Waltham, MA, USA), excitation/emission maxima of 490/516 nm.

### 2.4. Cell Growth Measurements

For cell growth experiments, HT-1080 fibrosarcoma cells were cultured in Corning cell culture flasks (surface area 25 cm^2^). Experiments used cell culture passages between 6 and 12 to prevent genomic abnormalities. For the cell counting assay, cells were seeded in flasks at a concentration of 125,000 cells per flask and incubated in 5% CO_2_ at 37 °C. Cells were allowed to proliferate exponentially in the presence of externally applied magnetic fields and were taken from the incubator at the end of the fourth day. The culture medium was removed, and the cells were washed with Dulbecco’s Phosphate-Buffered Saline (D-PBS) (ATCC 30-2200) (5 mL per flask). D-PBS was removed, and 1.5 mL of trypsin (ATCC PCS-999-004) was added per flask. The cells were placed in the incubator for five minutes. The culture flask was observed under a microscope to verify that the cells had detached from the bottom. Trypsin was neutralized by adding 4.5 mL of medium per flask. For human fibroblast cells, the same procedure was followed. However, to neutralize Trypsin, 1.5 mL of Trypsin neutralizing solution (ATCC PCS-999-004) and 3 mL of fibroblast growth medium were used. The solution was placed in a centrifuge tube and centrifuged at 2000 rpm for five minutes using the Centrifuge XC-2000 Premiere (C & A Scientific, Manassas, VA, USA). The supernatant was removed, and 1 mL of culture medium was added and gently mixed using a pipette to obtain a homogenous cell suspension. A 40 µL measure of the cell suspension and 40 µL of trypan blue were added and mixed gently several times. For each individual count, 10 µL was used.

In this study, we employed the Countess II Automated Cell Counter (Thermo Fisher Scientific, Waltham, MA, USA) to determine the cell count at the conclusion of the experiments. By utilizing this cell counter, we were able to obtain precise measurements of the cell population. Each cell growth experiment, involving different RF frequencies, was independently replicated twice (*N* = 2). In each experiment, we collected eight samples, resulting in a total of 16 samples. For each experimental condition, we calculated the mean and standard deviation based on the 16 measurements.

### 2.5. Statistical Analysis

Data were analyzed using Origin Pro 2023 Statistical Package (OriginLab, Northampton, MA, USA) with one-way ANOVA followed by Student’s *t*-test. All values are expressed as mean ± SD. The total sample count is denoted as ‘*n*’, while the number of independent replications is denoted as ‘*N*’. Statistical significance was considered at the following levels: * *p* < 0.05, ** *p* < 0.01, and *** *p* < 0.001. Data were presented where negative controls (untreated cells) normalized to 1.

## 3. Results

### 3.1. Oxidative Stress

Oxidative stress, characterized by an imbalance between the production of ROS and the cellular antioxidant defense system, plays a pivotal role in various physiological and pathological processes [18]. Excessive ROS generation can lead to cellular damage, including lipid peroxidation, protein oxidation, and DNA damage, ultimately impacting the cellular function and contributing to the development of numerous diseases [19]. To investigate and quantify oxidative stress levels in our study, we employed the fluorescence-based assay utilizing the CellROX Green reagent. CellROX Green is a cell-permeable dye that becomes fluorescent upon oxidation by ROS, allowing for the detection and measurement of intracellular oxidative stress levels.

As can be seen in Figure 4, we observed a significant increase in oxidative stress at 4.2 MHz in fibrosarcoma cells. Furthermore, we also found statistically significant results at 4 and 4.4 MHz, indicating a notable impact of the RF fields on oxidative stress levels in our experimental setup. Observed changes in fluorescence intensity are indicative of alterations in ROS, which are generated as metabolic byproducts of mitochondrial respiration chain. Intriguingly, our investigation revealed the highest decrease in oxidative stress levels, specifically 4.2 MHz in fibroblast cells (Figure 5), highlighting the frequency-dependent regulation of cellular redox homeostasis. Similar to fibrosarcoma cells, a statistically significant decrease in oxidative stress levels was observed at 4 MHz in fibroblast cells. The observed reduction in oxidative stress suggests a potential modulation of ROS generation and detoxification mechanisms at this frequency.

It is plausible that variations in hyperfine resonances across different cell types contribute to the distinct responses observed, potentially impacting the electron transport chain activity. At the genetic, molecular, and biochemical levels, various discernible distinctions have already been noted between the mitochondria of normal cells and cancer cells [20].

### 3.2. Mitochondrial Mass

Mitochondria, often referred to as the powerhouses of the cell, play a crucial role in cellular metabolism and energy production. The quantity and quality of mitochondria within cells, commonly referred to as mitochondrial mass, are important indicators of cellular health and metabolic activity. Changes in mitochondrial mass can reflect alterations in cellular energy demands, stress responses, or pathological conditions [21]. We utilized the MitoTracker Green FM probes, a widely employed fluorescent dye that selectively accumulates within active mitochondria, enabling precise quantification of mitochondrial mass in live cells. We report a novel association between RF fields and mitochondrial mass. Particularly around 4.2 MHz, RF fields increased the mitochondrial mass in fibrosarcoma cells significantly (Figure 6). HT-1080 cancer cells have more hyperpolarized mitochondria [22]. Hyperpolarized mitochondria can be a source of ROS [23]. Increased mitochondrial mass levels could be one of the reasons to neutralize increased ROS production.

In contrast to fibrosarcoma cells, changes in mitochondrial mass in fibroblast cells in response to RF exposure are less than 20% at max, and the lowest increase was observed at 4 MHz (Figure 7). This indicates that the metabolic rate response to RF is more limited in fibroblast cells as compared to fibrosarcoma cells.

### 3.3. Cell Growth

Cell growth rates play a vital role in understanding the overall health and functionality of cells. In our study, we measured cell growth rates for both fibrosarcoma and fibroblast cells, aiming to gain a comprehensive understanding of their responses to electromagnetic exposures. To encompass a wide range of cellular responses, we conducted experiments across the frequency spectrum between 3 MHz and 5 MHz. This frequency range was carefully chosen due to our anticipation of hyperfine resonances occurring within the iron sulphur clusters at these frequencies. By investigating this range, we sought to unravel potential resonant interactions between the electromagnetic fields and the iron sulphur clusters, which play a pivotal role in various cellular processes. In conjunction with assessing oxidative stress and mitochondrial mass, the analysis of cell growth rates provides a holistic perspective and allows us to investigate potential correlations and interplay between these key cellular parameters.

In fibrosarcoma cells, we observed a nonlinear response and cell growth was at its minimum at 4.2 MHz, where the oxidative stress peaked (Figure 8). This could indicate that excessive oxidative stress could lead to cellular damage and apoptosis, which could subsequently result in cell growth inhibition.

In fibroblast cells, cell growth measurements also demonstrated a non-linear response to the RF exposure frequencies (Figure 9) with significant increases in growth rates at 3.4, 4 and 5 MHz and significant inhibitions at 3 and 3.6 MHz. As opposed to oxidative stress, the fibroblast cell growth exhibits an inverse U-shaped pattern, with higher growth at intermediate frequencies and lower growth at both lower and higher frequencies between 3.6 and 4.4 MHz. This suggests a complex interplay between RF exposure frequencies and cellular responses. While error bars overlapping might be visually misleading, the statistical analysis allowed us to determine whether the observed differences are statistically significant.

## 4. Discussion

In the RF region between 1 MHz and 10 MHz, it is suggested that the mechanisms that determine magnetic effects on free radical reaction rates depend on quantum mechanical hyperfine resonances at magnetic nuclei of certain molecules rather than a magnetic interaction normally applicable to radicals formed from biological molecules [24].

This research advances the argument that metalloproteins within the electron transport chain of mitochondria play a crucial role in interacting with externally applied static and RF magnetic fields. These electron transport proteins contain transition metal ions, such as iron. Iron and other ferromagnetic materials have the ability to generate and maintain a magnetic field. Due to the presence of unpaired electrons, these metal ions possess a magnetic moment. Unpaired electrons maintain a magnetic moment approximately 600-fold greater than the magnetic moment of a proton [25]. The higher the number of unpaired electrons in an atom, the greater the potential for these electrons to align their spins with externally applied magnetic fields. In particular, the presence of unpaired electrons within iron sulphur clusters, with their hyperfine resonances ranging from 1 MHz to 10 MHz, makes them viable candidates for interacting with externally applied RF fields.

Given the chemical properties of iron–sulfur clusters, we applied RF magnetic fields ranging from 3 MHz to 5 MHz in combination with SMFs. Preliminary findings indicate that intracellular pH levels undergo changes within the frequency range of 1.8 MHz to 7.2 MHz [26]. It is possible that frequencies near hyperfine resonances disrupt the interconversion between the singlet and triplet states of these clusters. It is important to note that the specific parameters of SMF, including its intensity and orientation, can potentially influence the observed outcomes. In our case, we used an SMF with an intensity of 45 µT, oriented perpendicular to the growth plane.

The findings of our study indicate a significant impact of RF fields on oxidative stress levels in both fibrosarcoma and fibroblast cells. Previous literature has provided compelling evidence for the significant impact of RF fields on oxidative stress levels in various cell types. For instance, Usselman et al. conducted a study on rat pulmonary arterial smooth muscle cells (rPASMC) and demonstrated a frequency-dependent increase in H_2_O_2_ singlet state products, leading to cellular oxidative stress at 7 MHz [27]. Similarly, Castello et al. exposed fibrosarcoma HT-1080 cells to a combination of SMFs at 45 µT, oriented vertically to the growth plane, and weak 5 and 10 MHz RF magnetic fields at a 10 µT RMS intensity perpendicular to the static field. They observed a significant reduction in cell numbers by up to 30% on day 2 when cells were exposed to the combination of SMF and a 10 MHz RF magnetic field, compared to the SMF control cells. Moreover, cells exposed to 10 MHz RF magnetic fields for 8 hours exhibited a 55% increase in H_2_O_2_ production, indicating the presence of oxidative stress [28]. Likewise, Curley et al. investigated the effects of RF fields at a frequency of 13.56 MHz on human pancreatic cancer cells (AsPC-1 and Panc-1). Their study revealed impaired mitochondrial respiration and increased the production of ROS, signifying RF-induced stress on the mitochondria [29]. These collective findings across different cell types underscore the potential of RF fields to induce oxidative stress, which can lead to cellular damage and the development of diseases [30]. The variations in response among different cell types may be attributed to variances in hyperfine resonances and mitochondrial functionality. Overall, these studies provide substantial evidence linking RF fields to oxidative stress and its implications for cellular function and health.

Mitochondria, which are essential organelles within cells, play a critical role in cellular metabolism and energy production. The assessment of mitochondrial mass provides insights into cellular health and metabolic activity. Our study revealed a novel association between RF fields and mitochondrial mass. In fibrosarcoma cells, we observed a significant increase in mitochondrial mass, particularly around 4.2 MHz. The observed increase in mitochondrial mass is believed to be a compensatory mechanism aimed at counterbalancing the heightened production of ROS resulting from hyperpolarized mitochondria [31]. In cancer, mitochondrial functions are commonly altered, playing a direct role in metabolic reprogramming and conferring a high degree of adaptability to cancer cells in response to various stresses and the tumor microenvironment [32]. In contrast, the changes in mitochondrial mass in fibroblast cells were less pronounced, suggesting a more limited metabolic rate response to RF fields in these cells compared to fibrosarcoma cells.

Cell growth rates are indicative of cellular health and functionality. Our study investigated the impact of RF fields on cell growth rates in fibrosarcoma and fibroblast cells. The observed nonlinear response in cell growth rates, with the lowest growth rate at 4.2 MHz in fibrosarcoma cells, suggests that RF exposure can induce cellular responses that go beyond mere growth inhibition. The inhibition of cell growth at this frequency aligns with the peak oxidative stress levels, indicating a potential link between oxidative stress and cell death processes. The excessive oxidative stress induced by RF fields may lead to cellular damage and cell death, subsequently inhibiting cell growth. Apoptosis, a tightly regulated programmed cell death process, plays a crucial role in maintaining cellular homeostasis and eliminating damaged or unwanted cells [33]. Mitochondria, known to play a prominent role in apoptosis, are particularly affected by oxidative stress [34]. Numerous studies have demonstrated the role of ROS and oxidative stress in apoptotic pathways, highlighting their impact on cellular homeostasis and disease pathogenesis [35,36]. Under conditions of excessive oxidative stress, as induced by RF fields, cells can activate apoptotic mechanisms as a protective response to prevent the propagation of damaged cells. Additionally, mitochondrial dysfunction, triggered by oxidative stress, has been shown to play a prominent role in apoptotic cell death through the release of key apoptotic factors [37,38]. Therefore, it is reasonable to speculate that the observed inhibition of cell growth in response to RF treatment could be attributed to the induction of apoptotic cell death pathways, triggered by excessive oxidative stress and mitochondrial dysfunction. These findings highlight the importance of maintaining cellular redox balance and suggest a potential link between RF exposure, oxidative stress, and impaired cell growth.

In this paper, we proposed a mechanism by which externally applied RF fields can induce changes in mitochondrial chemistry by leveraging the hyperfine resonances of metalloproteins present in the electron transport chain. By utilizing the hyperfine resonant frequencies of these protein complexes, we were able to predict the appropriate RF ranges for our experiments conducted on HT-1080 human fibrosarcoma cells and dermal fibroblast cells. Cellular mechanisms underlying the observed effects of RF treatment were investigated. Our results demonstrate significant alterations in mitochondrial activity in fibrosarcoma cells upon RF treatment, leading to both increased and decreased activity levels. Notably, we observed distinct changes in mitochondrial activity at different frequencies of RF fields in fibroblast cells, highlighting the divergent responses between fibroblast and fibrosarcoma cells to RF treatment. These findings underscore the impact of RF fields on oxidative stress, mitochondrial mass, and cell growth rates in both fibrosarcoma and fibroblast cells. This study contributes to the growing body of knowledge concerning the biological effects of RF exposure and underscores the importance of further research in this field. By unraveling the intricate mechanisms underlying these effects, we can enhance our understanding of the potential risks associated with RF fields and explore strategies to mitigate their adverse consequences on cellular health. Comprehensive investigations are needed to fully comprehend the broader implications and potential applications of RF treatment.

## Figures and Tables

**Figure 1 biomolecules-13-01112-f001:**
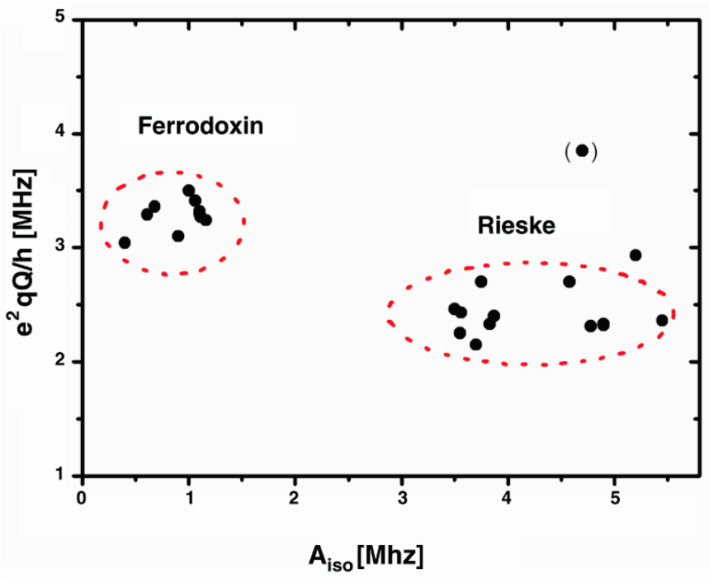
Plot of the hyperfine (A_iso_) and quadrupole (e^2^qQ/h) couplings of different iron sulphur clusters in different species [17].

**Figure 2 biomolecules-13-01112-f002:**
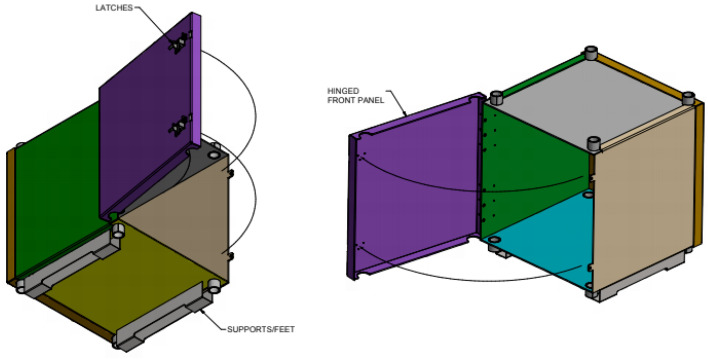
Mu metal box assembly.

**Figure 3 biomolecules-13-01112-f003:**
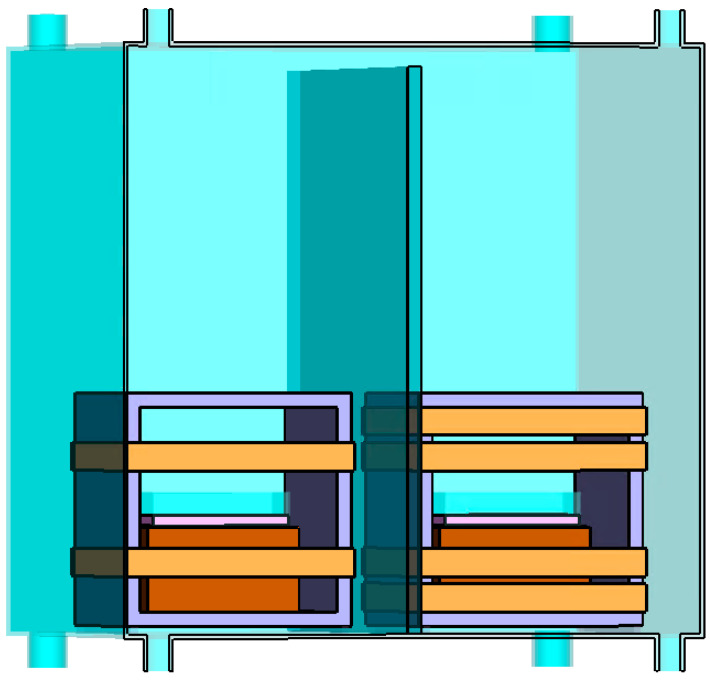
Experimental setup showing the configuration of two Helmholtz coils inside a mu-metal cage. The left coil consists of a single pair of turns and generates SMF for control cells. The right coil comprises two pairs of turns, the inner coil generates SMF, and the outer coil generates RF magnetic field for treated cells. A mu-metal sheet separates the two coils within the cage.

**Figure 4 biomolecules-13-01112-f004:**
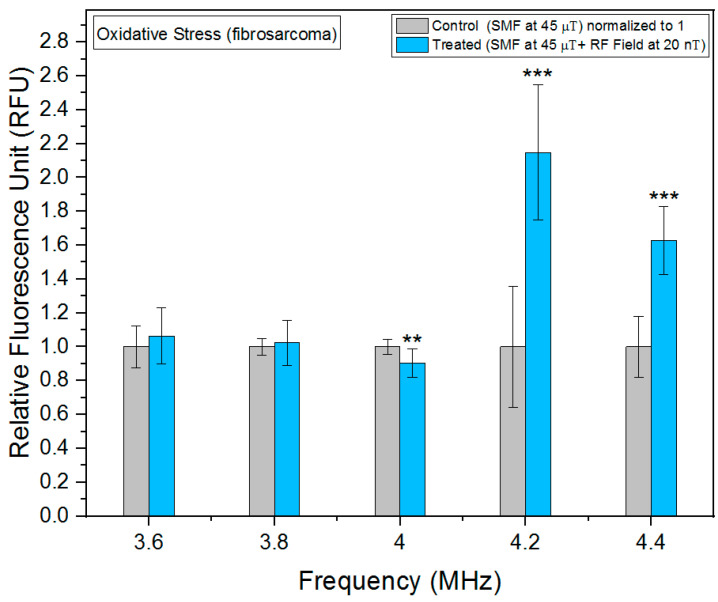
Oxidative stress as a function of frequency in fibrosarcoma cells. The data are expressed as mean ± SD; (*n* = 24, *N* = 2) for each group, and *p*-values are presented in Table A1 in Appendix A. ** *p* < 0.01, and *** *p* < 0.001 represent significant differences.

**Figure 5 biomolecules-13-01112-f005:**
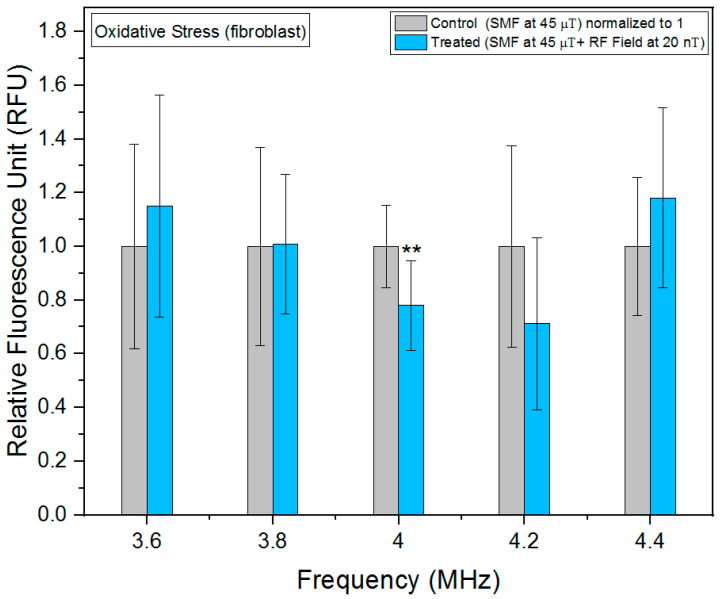
Oxidative stress as a function of frequency in fibroblast cells. The data are expressed as mean ± SD; (*n* = 24, *N* = 2) for each group, and *p*-values are presented in Table A2 in Appendix A. ** *p* < 0.01 represents significant difference.

**Figure 6 biomolecules-13-01112-f006:**
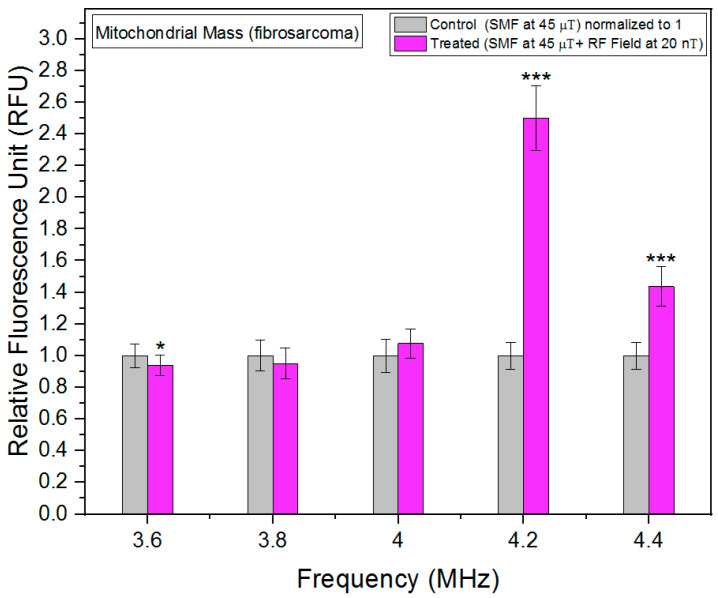
Mitochondrial mass as a function of frequency in fibrosarcoma cells. The data are expressed as mean ± SD; (*n* = 24, *N* = 2) for each group and *p*-values are presented in Table A3 in Appendix A. * *p* < 0.05, and *** *p* < 0.001 represent significant differences.

**Figure 7 biomolecules-13-01112-f007:**
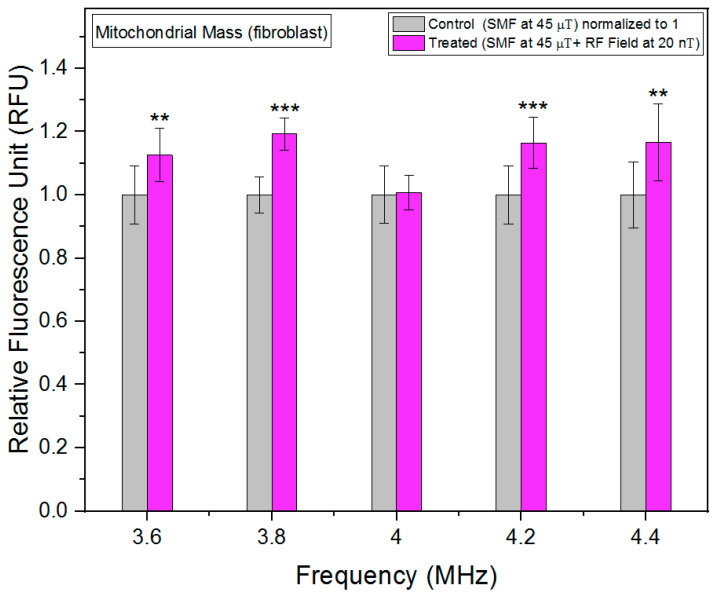
Mitochondrial mass as a function of frequency in fibroblast cells. The data are expressed as mean ± SD; (*n* = 24, *N* = 2) for each group, and *p*-values are presented in Table A4 in Appendix A. ** *p* < 0.01, and *** *p* < 0.001 represent significant differences.

**Figure 8 biomolecules-13-01112-f008:**
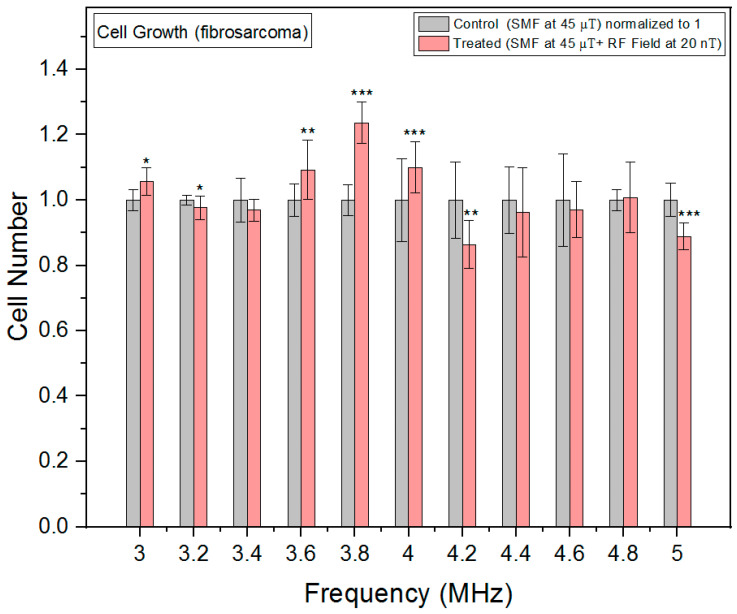
Cell growth rates as a function of frequency in fibrosarcoma cells. The data are expressed as mean ± SD; (*n* = 16, *N* = 2) for each group, and *p*-values are presented in Table A5 in Appendix A. * *p* < 0.05, ** *p* < 0.01, and *** *p* < 0.001 represent significant differences.

**Figure 9 biomolecules-13-01112-f009:**
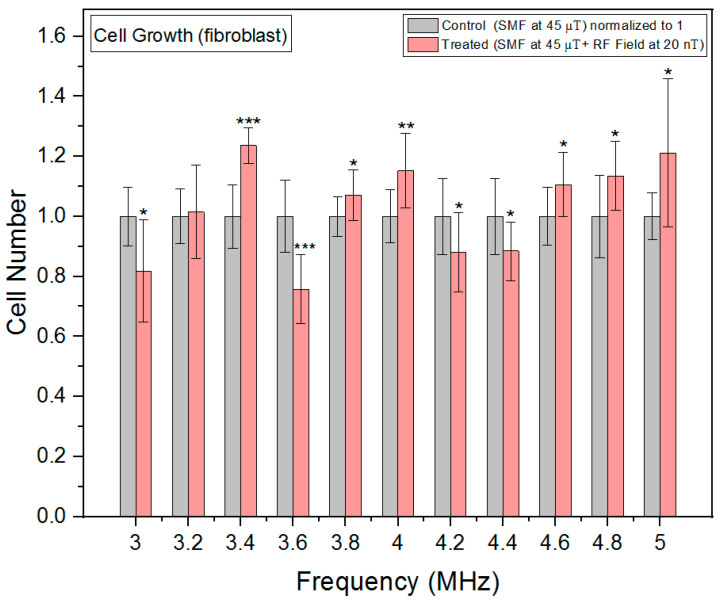
Cell growth rates as a function of frequency in fibroblast cells. The data are expressed as mean ± SD; (*n* = 16, *N* = 2) for each group and *p*-values are presented in Table A6 in Appendix A. * *p* < 0.05, ** *p* < 0.01, and *** *p* < 0.001 represent significant differences.

## Data Availability

The data presented in this study are available on request from the corresponding author.

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
