# Peer review of "Weak Radiofrequency Field Effects on Chemical Parameters That Characterize Oxidative Stress in Human Fibrosarcoma and Fibroblast Cells"

_biomolecules, 2023, doi:10.3390/biom13071112_

Round 1

Reviewer 1 Report

The results presented in this manuscript are very interesting and I think would be suitable for publication provided the following comments are address:

Should the title finish with a period? Please check journal style or consult with the Editor.

Abstract. The acronym “RF” is defined twice: once as “radiofrequency” and the second as “radio frequency”. Please define it only once, and be consistent throughout the manuscript.

Abstract. “We developed a model”… not sure what the authors mean with “model”. It looks more like they tested different exposures and reported the (very interesting, indeed) results. I see no model developed in this manuscript, but rather a mention of possibly connected phenomena that inspired the experiments. Please consider rephrasing. Alternatively, if they refer to a model presented in a previous work, please do let the reader of the Abstract know this.

Citation style. Page1 line 25: Shouldn’t “[1,2,3,4]” be “[1-4]”? Please check.

Introduction. Define “RF” before using it.

Page2.Line67. “We believe a resonant effect on the radical recombination can be influenced by background SMF via the electron Zeeman interaction.” The word “believe” sound odd in this context. The theory of radicals and their recombination, plus the Zeeman effect are extremely well document. Rather than declaring a belief the authors could explain how those theories combine to support what they are stating. Also, they could quantify what is the frequency/energy change due to a 45 µT SMF. And if that is their hypothesis, then it might be interested discussing the future possibility of repeating their experiments, but for different values of the SMF.

Page2.Line77. Fig1 shows that the range of hyperfine resonances is, roughly, 0.5-5.5 MHz, not 1-10 MHz. Please rephrase. Also, that figure is, as is well mentioned, from ref 15. The authors should reproduce it only with the corresponding permissions.

Materials and methods: In my opinion, the description in the text is not clear enough. Please add a scheme and/or a photograph of the whole set up, including the Helmholtz coils. Also state the number of turns of the coils and, ideally, their resistance and inductance values. What was the intensity of the current injected to the coils? Have you calculated the dissipated power by Joule effect? Further, the mu-metal box (thickness of their walls?) looks tight so, have you verified that the temperature, humidity and CO2 concentration is the same as outside of it? This is very important since even a small power can accumulate over the time of the exposure (four days in your experiments!) and eventually produce a shift in temperature. The authors state that “The RF magnetic field component between 3 MHz and 5 MHz was measured with a passive loop antenna”, but I could not find the value of the applied magnetic fields during the exposure in this section. The amplitude of 20 nT does appear in Fig. 3 legend… it should appear in the M&M section also, stating if those 20 nT are peak, peak-to-peak, or RMS value. I assume the amplitude of the RF fields was the same for all frequencies tested. If so, please state it explicitly. Also, please indicate the spatial direction of the 45 µT SMF and that of the RF MF (both vertical, horizontal,…?). Also what the attenuation produced by the mu-metal box: what were the DC and AC background fields inside (at the expose and at the control plates) with the coils unenergized?.

At the MHz range, the induced electric fields might become not-negligible. Please provide an estimation of those fields. Could have been local temperature rise due to specific absorption rate (SAR) by the cell culture? Please do cite pertinent reference(s) in this regard.

Lines 47-48 are identical to 73-74. Amend.

Lines 49-51 are identical to 70-72. Amend.

Page 4, lines 137-138. The authors state: “A total of 24 measurements were taken per parameter and therefore for each experimental condition, the mean and standard deviation corresponding to 24 measurements were presented”. This confuses me: there were six treated wells (and six controls) per experiment (right?), then how come there are 24 readings? Were, perhaps, 4 readings per well taken? If so, why? Also, and most importantly, if 24 readings were taken in each experiment, and the results are mean +/- SD with n = 24, does this mean that only one experiment was performed for each frequency? Please clarify, stating explicitly how many times the experiments were repeated (may be you performed each experiment four times and that is where the 24 comes from?). Also, please add the number of experiments in all the figures’ captions.

Four days might be a long time for a culture to go by without refreshing the culture medium... did you notice a change in the medium color, suggesting change in the pH? Depletion of nutrients and accumulation of metabolic detritus represent by themselves a condition of stress. I think that the manuscript would benefit if the authors discuss this, noting that the stress by the RF fields was concomitant to the stress implied by the long culture time. It would be interesting (perhaps for a future work) to change the medium by new, fresh medium at day 2 of exposure).

Page5, lines 193-211. These paragraphs seem to belong more to an introduction/discussion, rather than to the Results section. Please consider relocating them somewhere else in the manuscript.

Page5-6, lines 213-216. Same comment as previous: this sounds like introductory/discussion content, rather than results. Also, references 19 and 20 in these lines refer to ELF studies, not RF; thus they are hardly pertinent to this manuscript. Were the authors to decide keeping them, please let the reader know that they refer to ELF, not RF experiments. Preferably, the authors could replace them with RF studies.

In my opinion, the red continuous lines in Figs 3 and 5 are hardly justified. For instance, in Fig. 5 the red line passes way below the mean (even below the mean minus the SD), so it adds little, if something to the data. Further, it could mislead the reader by suggesting a tendency not dictated by the data. I suggest removing those red lines. Were the authors to decide keeping them, please make explicit what kind of regression they used, which were their parameters, and how good was the fitting (R2 value).

Page 56, lines 221-2. “We believe these changes are associated with changes in concentration of ROS as they are generated as metabolic byproducts of mitochondrial respiration chain”. This statement needs revising/rephrasing or deleting, since it seems like a hypothesis of the authors (“We believe…”) when it is actually well known that “CellROX Green is a fluorogenic probe used for measuring oxidative stress in live cells. The dye is weakly fluorescent inside cells in a reduced state but exhibits bright green photostable fluorescence upon oxidation by ROS and subsequent binding to DNA.” (Lozoya-Gloria et al., Bio Protoc. 2017 Aug 20; 7(16): e2508).

The authors don’t make any mention to the statistically significant results of Fig. 3 at 4 and 4.4 MHZ. Please do.

Lines 223-224. The authors state “On the other hand, we observed a decrease at 4.2 MHz in fibroblast cells (Figure 4). This might be due to different cells have different hyperfine resonances which may affect the electron transport chain activity at different frequencies.” This is too lightly dropped on the reader… Also the way it is written suggests that cells have hyperfine frequencies when it is atoms/molecules that have them… Are the authors suggesting that the electron chain is different in those two different cell lines… could they produce one (or more) references supporting such a hypothesis? Moreover, the decrease seen at 4.2 MHz is not statistically significant so it should not be discussed as it were… maybe the authors wanted to write “4 MHz” (statistically significant) instead of “4.2 MHz”? Also they do not mention that both cell lines showed a significant decrease at 4 MHz. This might be worth discussing.

Line 230. “Oxidative stress induced cellular damage and changes in mitochondrial physiology are considered to lead apoptosis [21].” Why do the authors bring this up here in the manuscript? They are not discussing apoptosis, nor did they measured it… perhaps it would be suitable when discussing their cell growth results, or in the Discussion.

Line 236. “It’s worth noting that RF fields induced similar characteristics on the levels of oxidative stress.” I don’t follow: Figs 3 and 4 show remarkable differences. Please revise/amend.

Line 251: MHZ should be MHz (lowercase z).

Lines 250-252. “Lastly, we investigated the RF effects on cell growth in fibrosarcoma cells between 3 MHz and 5 MHZ with focus on frequencies around 4 MHz due to biggest changes in oxidative stress response in that range”. I don’t follow: why do the authors state “due to biggest changes in oxidative stress response in that range”? That is the sole range studied. The statement suggests that a wider range was studied in oxidative stress experiments, and that 3-5MHz showed the biggest effects, and that is not the case. On the contrary, oxidative stress was studied in a narrower range (3.6-4.4 MHz). Alternatively, if they mean that they studied growth of fibrosarcoma cells only, because those cells showed a bigger effect than fibroblasts, then they should state that. Please revise/amend.

Lines 253-4. “This could indicate that excess amount of oxidative stress at 4.2 MHz could be triggered apoptosis.” Please revise phrasing/grammar.

Line 264: “indicate” instead of “indicates”?

In my opinion, the Discussion section could be benefited by the inclusion of references to studies somehow similar to the ones performed by the authors. This would give context to their results. Literature provides abundant examples of RF and oxidative stress in in vitro studies. Also, for instance, they could mention that their results could depend on the chosen value (and direction) of the SMF (45 µT). Also, they could cite examples of studies that tested different frequencies and also found a frequency-dependent effect (for the same endpoints studied in the manuscript, or others). Etc… In its present state the Discussion seems more like a summary. I suggest enriching it with some context from the state of the art.

Only minor corrections, see comments above.

Reviewer 2 Report

In this study the authors had two cell lines undergo weak radiofrequency (RF) fields (a human fibrosarcoma cell line and a human primary fibroblast cell line). Although as stated in the introduction “the objective of our work is to improve our understanding of the mechanisms by which weak electromagnetic fields modify chemical reactions rates that in turn change processes in biological system and lead to health effects” [line 25-27], there are no real evidence or experiments able to illustrate any mechanism by which RF can influence any aspects of biological systems. In this respect, even the title of the manuscript appears to be misleading and inappropriate: in fact, there are no analysis of “chemical parameters that characterize oxidative stress” presented here. Along the same line, even in the abstract section the authors state that “This work aims to improve our understanding of how externally applied weak radiofrequency fields alter concentrations of chemical parameters that characterize oxidative stress” [line 9-10]: actually, there are no experimental evidence presented here that are able to explain “how” RF can alter biological processes.

In all, this study essentially shows three analyses conducted after four days of RF exposure: an oxidative stress and a mitochondrial analysis based on fluorescent quantifications and a cell count analysis. These analyses alone are not sufficient to describe any plausible mechanism related to the effect of RF on the treated cells. In the current form this study appears to be barely descriptive, without any mechanistic insight, poorly written and therefore, not suitable for publication.

Specific comments:

1)    The authors used cells kept with a magnetic field of 45 μT as control (SMF) that were compared to cells treated with RF (SMF + RF). Cells kept in normal conditions (in a separate incubator and not in a Mu metal box) should be included as a control condition as well. In the ‘Materials and Methods’ section [line 132-134], the authors mention to be using control cells without magnetic field exposure but there are no data about those control cells. Moreover, it seems (but it is not very clear) that the data presented here are derived from 24 measurements of one single experiment [line 136-139]. If this is correct, there is a need to perform replicates of the experiments.

2)    All the experiments were performed after 4 days of exposure to RF. One concern is that RF may influence apoptosis and/or autophagy. Especially in HT-1080 cells that are known to be genetically heterogeneous and genetically unstable (the ATCC website reports 40% of HT-1080 cells to have rearranged karyotypes), it could be possible that a subset of cells might be more vulnerable to RF and more prone to apoptosis or autophagy. This may cause a sort of selection for RF-resistant cells, a sub-population with different properties selected during 4 days of RF exposure (different ROS production, different number or dimension of mitochondria, a slightly different growth rate). Apoptosis and/or autophagy should be monitored during RF treatment (including time points during the RF treatment up to 4 days) to test this hypothesis and to clarify whether apoptotic/autophagic events are induced by RF exposure in the cells used in this study.

3)    In all the histogram graphs from Fig. 3 to Fig.6 there is a curvy line connecting the histograms that is not described in the text or in the figure legends.

4)    A point of concern is the fact that the values of the histograms and of the error bars are not indicated. In some of those graphs, error bars are very overlapping and even if the authors indicate those values as statistically significant (as in Fig.4, point of frequency 4 MHz, or in Fig.5, point of frequency 3.6 MHz, or in Fig.7), it is not very convincing. A table with values for all the figures should be submitted as supplementary data.

5)    What the authors indicate as Mitochondrial Mass (Fig.5 and Fig.6) derives from the quantification of the total fluorescence value of the mitochondrial dye ‘mitotracker’. An increase in fluorescence could be due to an increasing number of mitochondria or could be due to an increased mitochondrial surface. The authors did not investigate this aspect. Increasing ROS values suggest mitochondrial damage: EM ultrastructure images of those cells would help to understand whether the mitochondria structure is altered.

6)    Cell count analysis of the fibroblast cell line has not been performed.

7)    The ‘Discussion’ section of the manuscript is very incomplete (almost non-existent). The author’s findings should be discussed considering the existing literature, highlighting points of strength and limitations of the study.

Reviewer 3 Report

This article uses sound techniques both in biological techniques and in setting up exposures to assess the effect of MHz magnetic fields on cellular impacts. The numerous biological techniques, growth, fluorescence, oxidative stress  and mitochondrial mass used to document the effect increases confidence in the reliability of the results.

The contribution here is to confirm that continuous signals have effects on biological processes, although it is widely believed that the modulation of such RF-MW signals, mostly at lower frequencies, is most strongly bioactive. The results of this study are expected because biological structures are diverse enough to display resonances at a wide variety of frequencies, probably from low ELF to beyond the GHz range.

Consideration of both electron and proton effects by RF radiation credits the analysis.

The field value used in the experiments, 20 nT, is not prominent in the paper, and some will wonder if that field intensity, which roughly corresponds to radiated 100,000 µW/m², is a non-thermal value.

Just a few minor typos.

Reviewer 4 Report

The manuscript is well-written but some minor changes are needed.

Supplementing the text with examples will make it easier for the reader who is not proficient in the subject to understand the work.

·       Line23 – “Relatively weak magnetic fields…”- be specific, provide example

·       Line 27-31- “There can be a long chain of  events that lead from the physics of an electromagnetic field…” Isn't that low-frequency EMF action? Won't thermal effects dominate at MHz?

·       Line 35- “The fact that radiofrequency magnetic fields can affect biological systems is well established…”

·       Line 36-41 “Consistent with the expanding number of studies, the work presented in this paper includes experimental results on the effects of RF fields on both fibrosarcoma and fibroblast cells and proposes a mechanism by which an RF magnetic field at nano-Tesla (nT) intensity can alter the growth rates, oxidative stress and mitochondrial mass based on previously discovered phenomenon of “Radical  Pair Mechanism”. “ -The sentence is too long and it's easy to lose the thread. It's good to add some example of the action of EMF of n-Tesla based (what frequency?) on the phenomenon of “radical pairs” (for a better understanding of the assumptions).

·       Line 97- static magnetic fields- maybe it is worth adding the abbreviation SMF in the further text, so that the reader has no doubts that it is a static magnetic field (instead of “magnetic fields”)

·       Fig. 2 Instead of a Mu metal box drawing, a scheme of “SMF and RF generators” arrangement in the incubator would be more useful.

·       Line 132- “Six wells were treated with magnetic field (SMF = 45 µT) and six were taken as control.” - I am confused: “The control cells in Helmholtz coils were maintained with a magnetic field of 45 µT and treated cells were exposed to different flux densities of magnetic fields (Line 105)”.

·       The experimental protocol including SMF and RF exposure (when, what field, what values; scheme or description of steps) is definitely missing.

·       Edit: After reading the whole text and the markings on the charts, the question of the SMFi RF exposure became clear. I think the description in the method section (line 105) should be changed: "The control cells in Helmholtz coils were maintained with a magnetic field of 45 µT and treated cells were exposed to different flux densities of magnetic fields." This suggests that SMF will also be changed. I suggest using the abbreviation SMF and RF. In addition, write clearly that Helmholtz coils were used to generate SMF with a "background" value - 45 uT (earth magnetic field; shielded by a mu-metal box and incubator).

·       Line 199 and 208 “externally applied static and RF magnetic fields.” The exposure concerns a constant magnetic field with a background value (natural conditions). It's rather inappropriate to talk about a combination of two fields.

·       The description of the number of cells and/or repetitions, and exposure time is chaotic and not clear. It is worth adding such information in e.g. the caption of the results figures.

Round 2

Reviewer 1 Report

Dear authors,

Upon a substantial improvement of your manuscript, I am recommending acceptance provided you consider taking care of these two final observations:

I still think it would make it easier for the reader that the authors state: "independent experiments were performed twice" or "in duplicate". The number "2" is still missing from the M&M section, forcing the reader to do the math from "24" (mentioned several times) to "2". The phrase "repeated on separate days" is, in my opinion, not completely free of ambiguity. On the same line, figure captions are still missing the number of independent replications of the experiments, which is 2: N = 2. This, in my opinion, is necessary: both in the M&M section and in the Figure captions: n = 16 (or 24), and N = 2.

Also, in the Tables, I suggest changing the title for the first column from "Frequency" to "Frequency (MHz)" and deleting "MHz" in each of the rows, which is clearly redundant.

Best regards.

Reviewer 2 Report

The authors have provided a new revised version of the manuscript, but the quality of the study has not been substantially improved. Most of the major concerns expressed during the first revision process still remain.

The issue about a missing mechanism through which RF can influence biological systems is still completely unresolved. The revised manuscript does not provide additional experiments to improve the understanding of “how” RF act. The study remains descriptive: the effect of RF on oxidative stress, mitochondrial mass and cell growth are observed but there is no explanation, no mechanism, or logic connection between these phenomena. For example, by looking at the oxidative stress in fibroblasts, it appears that only at frequency of 4 MHz there is a significant change. But when looking at the mitochondrial mass, at frequency of 4 MHz there is no significant change between control and treated samples. On the other hand, significant changes in mitochondrial mass are observed at 3.6, 3.8, 4.2 and 4.4 MHz where paradoxically no changes in oxidative stress are detected. How do the authors explain those discrepancies? It is difficult to establish a logical connection between the data presented here. Same goes for the cell growth rates where it is possible to notice that for example at 3.4 MHz fibroblasts are growing significantly more, at 3.6 MHz they grow significantly less, and then at 3.8 MHz they again grow significantly more, while no significant oxidative stress is detected at those frequencies. It is hard to explain or find a correlation or a causal connection between the data.

I continue to believe that the title of the study is inappropriate since it is not clear which “effects on chemical parameters that characterize oxidative stress” the authors are referring to. There are no chemical parameters analyzed here but only oxidative stress (in a very general way by using a kit detecting ‘oxidative stress’), mitochondrial mass and cell growth: these are not chemical parameters.

Also, data from cells kept in normal condition (in a normal incubator without Mu metal box) have not been added to the study (Point 5 in the response letter). This is an important control that is still missing.

The issue about apoptosis (point 6) has not been addressed. A change in apoptosis during the 4 days of RF exposure could help explaining if RF would affect only a subset of susceptible cells and so help to explain the oscillating and puzzling cell growth changes observed in the study.

Also, no additional experiments have been done to clarify whether the mitochondrial mass change is due to an increase in the total number of mitochondria or if the mitochondria become bigger (point 9).

In conclusion, I don’t find this study to be significantly improved from its original submission.
